# Immunoreactivity of a Putative ECF σ Factor, LIC_10559, from *Leptospira interrogans* with Sera from *Leptospira*-Infected Animals

**DOI:** 10.3390/pathogens12040512

**Published:** 2023-03-25

**Authors:** Sabina Kędzierska-Mieszkowska, Zbigniew Arent

**Affiliations:** 1Department of General and Medical Biochemistry, Faculty of Biology, University of Gdańsk, 80-308 Gdańsk, Poland; 2University Centre of Veterinary Medicine UAK, University of Agriculture in Krakow, 30-059 Krakow, Poland; zbigniew.arent@urk.edu.pl

**Keywords:** immunoreactivity, infection, *Leptospira*, leptospirosis, pathogen, sigma factor

## Abstract

*L. interrogans* belongs to highly invasive spirochaetes causing leptospirosis in mammals, including humans. During infection, this pathogen is exposed to various stressors, and therefore, it must reprogram its gene expression to survive in the host and establish infection in a short duration of time. Host adaptation is possible thanks to molecular responses where appropriate regulators and signal transduction systems participate. Among the bacterial regulators, there are σ factors, including ECF (extracytoplasmic function) σ factors. The *L. interrogans* genome encodes 11 putative ECF σ^E^-type factors. Currently, none of them has been characterized biochemically, and their functions are still unknown. One of them, LIC_10559, is the most likely to be active during infection because it is only found in the highly pathogenic *Leptospira*. The aim of this study was to achieve LIC_10559 overexpression to answer the question whether it may be a target of the humoral immune response during leptospiral infections. The immunoreactivity of the recombinant LIC_10559 was evaluated by SDS-PAGE, ECL Western blotting and ELISA assay using sera collected from *Leptospira*-infected animals and uninfected healthy controls. We found that LIC_10559 was recognized by IgG antibodies from the sera of infected animals and is, therefore, able to induce the host’s immune response to pathogenic *Leptospira*. This result suggests the involvement of LIC_10559 in the pathogenesis of leptospirosis.

## 1. Introduction

*Leptospira interrogans* is the main causative agent of leptospirosis, which is one of the most common zoonotic diseases worldwide [1]. It is estimated that there are over one million cases of human leptospirosis and approximately sixty thousand deaths due to this disease annually [2]. Humans can become infected with *Leptospira* through direct contact with the urine of infected animals or indirectly through contact with urine-contaminated water or moist soil [3]. However, since most human cases originate from soil or water contaminations, leptospirosis is also considered an environment-borne infection [4]. It should be remembered that leptospirosis is also a serious economic problem, because it causes abortions, stillbirths, infertility, failure to thrive, reduced milk production and death in farm animals, such as cows, pigs, sheep, goats and horses [5,6,7]. Thus, leptospirosis is a serious disease that has a significant effect on both public health and the economy worldwide. Despite its global importance and severity, the molecular mechanisms of this disease’s pathogenesis, as well as leptospiral virulence, are still largely unknown.

During its complex life cycle, *L. interrogans* is exposed to various environmental stimuli, and in order to survive unfavorable conditions and adapt to the selected host, this pathogen must reprogram its gene expression. Indeed, numerous studies of *L. interrogans*’ global transcriptional responses to various environmental signals, such as temperature and pH changes, iron limitation, osmolarity and so-called host-induced stresses (involving interaction with phagocytic cells and the complement system activation), have shown that this pathogen alters the expression of many of its genes [8,9,10,11,12,13,14,15]. The genes associated with environmental and host adaptation, which ensure *Leptospira*’s survival, are upregulated. Among them, there are genes encoding virulence factors that are necessary for the mentioned successful host adaptation and establishment of infection in a short duration of time [16]. Thanks to these virulence factors, the pathogen can invade the host and evade its host’s immune response during the initial phase of infection. Currently, there is limited knowledge of the gene regulation in *Leptospira*, including virulence gene factors, and revealing their regulation is necessary for understanding the molecular basis of leptospirosis or developing effective diagnostic methods and strategies for combating this pathogen.

Transcription, which is the initial step in gene expression regulation, is often determined by environmental changes, including the host-induced stress. Therefore, it is no surprise that transcriptional regulation and σ factors, involved in this process by controlling the promoter selectivity of bacterial RNA polymerase, play a key role in the environmental/host-induced gene expression response in bacteria, including pathogenic *L. interrogans.* Recent large-scale genomics and bioinformatics analyses revealed that the *L. interrogans* genome, consisting of 2 circular chromosomes, encodes a total of 14 putative σ factors [17,18], namely, the primary σ^70^ factor (LIC_11701, also referred to as RpoD); σ^28^ (LIC_11380, similar to σ^F^, FliA); and 11 ECF σ factors (σ^E^-type), which belong to the σ^70^-family; and one σ^54^ (LIC_11545, also known as RpoN) (Table 1) that represents the σ^54^-family, functionally similar to, but structurally distinct from, σ^70^. Unfortunately, none of these factors have been biochemically characterized so far, and their functions are not fully known. The most mysterious group of leptospiral σ factors are ECF (extracytoplasmic function) σ factors (ECF σs). In other bacterial species, ECF σs mediate the envelope stress response, contribute to heat shock and oxidative stress responses, and are involved in the regulation of virulence genes in many bacterial pathogens [19,20,21,22,23,24]. Thus, these factors are important signal-response regulators in the environmental stress responses. Among leptospiral ECF σs, there is LIC_10559 detected only in the highly pathogenic *Leptospira* [17]; therefore, it is speculated that this factor is active during infection.

This study provides the first insight into the role of LIC_10559 during leptospiral infection. We achieved overexpression of LIC_10559 in the *E. coli* pET system to evaluate its immunogenic potential, which could point to this factor’s role in the pathogenesis of leptospirosis. We demonstrated that LIC_10559 is able to induce an antibody response in animals infected with *Leptospira.* Recognition of LIC_10559 by the humoral immune response to *Leptospira* strongly supports speculation that this protein is active during infection.

## 2. Materials and Methods

### 2.1. Serum Samples

In this study, we used archived serum samples from rabbits (*n* = 8) and cattle (*n* = 10). Rabbit antisera against *L. interrogans* serovars (Icterohaemorrhagiae, Hardjo and Canicola) and *L. borgpetersenii* serovars (Hardjo and Javanica) were prepared as described by [26]. A rabbit pre-immune serum was used as a negative control. Bovine sera were collected from cattle experimentally infected with the *L. borgpetersenii* serovar Hardjo via conjunctival instillation of 1 × 10^6^ bacteria. Blood samples were collected 28 days after the challenge and, in one case, 210 days after the challenge. Sera from uninfected cattle (*n* = 7), as well as fetal bovine serum, were used as negative controls. To confirm the serological status of leptospiral infection, the sera were subjected to a microscopic agglutination test (MAT) [26,27] and used at dilutions 1:100 for Western blotting or 1:200 for ELISA assay.

The sera used in this study were originally collected for another study (project license number PPL2608, date of approval 15 October 2008). All operators involved in the study, protocols and premises were licensed under the Animals (Scientific Procedures) Act (1986) (ASPA).

### 2.2. Plasmid Construction for Protein Overproduction

The *L. interrogans LIC_10559* gene (546 bp) was amplified from the genomic DNA of the *L. interrogans* serovar Copenhageni by PCR using Pfu DNA polymerase (EURx, Gdańsk, Poland) with the following primers: CATATGATGACTGAATCCGAATTTGC, with the NdeI restriction site underlined, and AAGCTTTCATTCTTCATAAAATTTCTCC, with the HindIII restriction site underlined. DNA primers were synthesized by Sigma-Aldrich (Merck KGaA, Darmstadt, Germany). The PCR reaction mixture (50 μL) was composed of 1x Pfu buffer containing 1.5 mM MgSO_4_, 0.2 mM of each dNTP, 0.2 mM primers, 200 ng of genomic DNA and 2.5 units of Pfu DNA polymerase and nuclease-free water added to a final volume of 50 μL. The PCR reaction was carried out in a Bio-Rad MJ Mini Personal Thermal Cycle PTC-1148 under the following conditions: initial denaturation at 95 °C for 5 min, 95 °C for 30 s, annealing at 59 °C for 30 s, elongation at 72 °C for 35 s, and a final elongation at 72 °C for 4 min. The reaction was carried out for 35 cycles. For evaluation of the PCR reaction, the PCR product (546 bp) was electrophoretically resolved in a 1% agarose gel and visualized in the presence of ethidium bromide under UV light. The PCR product was purified using a Clean-up kit (A&A Biotechnology, Gdańsk, Poland) and used for cloning into a pJET1.2 blunt vector (ThermoFisher Scientific, Warsaw, Poland), then digested with NdeI and HindIII (New England Biolabs, Ipswich, MA, USA), and ligated with the linearized pET28 NdeI-HindIII vector. The sequence of the resulting construct was confirmed by DNA sequencing (Genomed S.A., Warsaw, Poland). *Leptospira* genomic DNA was extracted with a QIAamp DNA Mini Kit (Qiagen, Wroclaw, Poland). The DNA plasmid preparation and transformation of *E. coli* cells were done according to [28].

### 2.3. Fractionation of E. coli Cells—Extraction of the Recombinant LIC_10559 Protein

For fractionation, 10 mL cultures of *E. coli BL21* (λDE3) bacteria transformed with the recombinant plasmid pET28 carrying *LIC_10559* were used. For this purpose, the bacteria were grown in the presence of kanamycin (30 μg/mL) at 28 °C or 37 °C to OD_600_ ~0.5, and then IPTG (0.4 mM) was added to induce *LIC_10559* expression. After a 2 h IPTG induction, bacterial cultures were collected by centrifugation (5000× *g* for 15 min), resuspended in 0.5 mL of CelLytic B Bacterial Cell Lysis Extraction Reagent (10× diluted) (Sigma, St. Louis, MO, USA) and then shaken for 15 min. Then, the bacterial suspension was centrifuged (15,000× *g* for 10 min at 4 °C), and the obtained supernatant (soluble fraction) was transferred into a new tube, while the remaining pellet was resuspended again in 0.5 mL CelLytic reagent (10× diluted), and lysozyme was added to a final concentration of 0.2 mg/mL. This extraction suspension was incubated for 15 min with shaking, 25 units of Benzonase nuclease (Sigma) was added, and the incubation was continued for another 15 min. Next, the suspension was centrifuged (15,000× *g* for 15 min), and the second supernatant (an additional fraction obtained after lysozyme addition) was collected, while the pellet (insoluble fraction) containing the protein aggregates (inclusion bodies) was resuspended in 100 µL of the Laemmli lysis buffer. All fractions were analyzed by SDS-PAGE electrophoresis and Coomassie blue staining.

### 2.4. Purification of the Recombinant LIC_10559 Protein

*L. interrogans* LIC_10559 protein was overproduced as a hexa-histidine fusion protein in an *E. coli* BL21(λDE3) strain (Novagen by Merck KGaA) and then purified using immobilized metal affinity chromatography (IMAC) and a Protino Ni-NTA Agarose (Machery-Nagel GmbH & Co.KG, Düren, Germany). A general his-tagged protein purification procedure was employed with a few modifications. CelLytic B Bacterial Cell Lysis Extraction Reagent (Sigma St. Louis, MO, USA) was used for the lysis of bacterial cells. The wash buffers contained 50 mM Tris-HCl (pH 8.0), 300 mM NaCl, 5–60 mM imidazole (Sigma-Aldrich), 20% glycerol (Sigma-Aldrich) and also 0.1% Triton X-100 (Sigma-Aldrich), while the elution buffers contained an increased concentration of imidazole, i.e., 100 and 250 mM. Bacteria were grown in the presence of kanamycin (30 μg/mL) at 37 °C to OD_600_ ~0.6 (app. 6 h), and then IPTG (0.4 mM) was added to induce *LIC_10559* expression, and the cells were grown further for 4 h at 28 °C. Then, bacterial cultures were harvested by centrifugation at 7000× *g* for 20 min at 4 °C (~8 g wet cells were obtained from 2 L of bacterial culture), and suspended in 24 mL of CelLytic reagent supplemented with lysozyme (1 mg/mL) and 250 units of Benzonase nuclease (Sigma). The bacterial suspension was then incubated on ice with shaking for 30 min, and disrupted by sonication (5 pulses of 30 s with breaks of 30 s after each pulse, amplitude 20%, using a Vibra-Cell sonicator) in the presence of the protease inhibitor PMSF (1 mM) and centrifuged (20,000× *g*, 1 h). The supernatant was applied onto a column packed with a Protino Ni-NTA agarose, and the bound protein was eluted with 100 and 250 mM imidazole. Fractions containing his_6_-tagged LIC_10559 (calculated molecular mass of 23,381 Da) were analyzed by SDS-PAGE electrophoresis and Coomassie blue staining. The pooled fractions containing LIC_10559 were dialyzed against dialysis buffer consisting of 20 mM Tris-HCl pH 8.0, 500 mM NaCl, 20% glycerol, 1 mM EDTA and 1 mM DTT. Glycerol was then added to a final concentration of 50%, and the purified protein was stored at −20 or −70 °C. The protein concentration was estimated by the Bradford method, using Bradford reagent (BioRad, Warsaw, Poland), with BSA as a standard.

The identity of the recombinant LIC_10559 protein was confirmed by a liquid chromatography-tandem mass spectrometry (LC-MS/MS) analysis of the tryptic peptides obtained after trypsin cleavage of the protein, performed at the MS LAB IBB PAN (Warsaw, Poland). The equipment used was sponsored in part by the Centre for Preclinical Research and Technology (CePT), a project co-sponsored by the European Regional Development Fund and Innovative Economy, The National Cohesion Strategy of Poland.

### 2.5. SDS-PAGE and Western Blotting Analysis

To assess the immunoreactivity of LIC_10559, SDS-PAGE electrophoresis (using 12.5% or 15% polyacrylamide gels) and Western blotting with ECL (Enhanced Chemiluminescence) detection were performed. The blots were blocked with 0.1% Tween 20 in Tris-buffered saline (TBS) for 1 h at room temperature and then incubated overnight at 4 °C with polyclonal rabbit or bovine sera (1:100 dilution) against *Leptospira* strains. After primary antibody incubation, the blots were washed three times with TBS containing 0.05% Tween 20 and incubated for 1 h at room temperature with the goat anti-rabbit IgG horseradish peroxidase (HRP) conjugate (Abcam) diluted 1:3000 or the polyclonal rabbit anti-cow Ig/HRP conjugate (BioRad, Poland) diluted 1:1000. The blots were then washed three times as described above and were developed using BioVision ECL substrates (Gentaur, Sopot, Poland) and imaged using the Azure imaging system (Azure Biosystems, Dublin, CA, USA).

### 2.6. ELISA Assay

ELISA (enzyme-linked immunosorbent assay) was performed to analyze the immune response in animals experimentally exposed to *L. interrogans* serovars, according to the procedure described previously [29]. Briefly, Costar 96-well EIA/RIA polystyrene high-binding plates were coated with 100 μL of 0.313 μg/mL of the recombinant LIC_10559 (a capture antigen) resuspended in phosphate-buffered saline (PBS) by incubation overnight at 4 °C. Control and duplicate bovine serum samples were diluted 200-fold in PBST buffer; the diluted sera were applied to each well in duplicate. HRP-conjugated anti-cow IgG (BioRad Poland) (diluted 1:4000) was used as a secondary antibody. The reaction was stopped after 10 min by the addition of 50 μL of 1 M H_2_SO_4_. The absorbance at 450 nm was measured using a PerkinElmer Multimode Plate Reader (Enspire, Waltham, MA, USA). The assay was performed in triplicate for each serum.

### 2.7. Data Analysis

All the graphs and statistical analyses were performed using GraphPad Prism software. To test whether the collected numerical data are normally distributed, the Shapiro–Wilk normality test was applied. Statistically significant differences between the ELISA results obtained for sera collected from uninfected and infected animals were determined using the Mann–Whitney U test. *p* < 0.05 was considered statistically significant. The results of the data analyses are presented in the graphs as the median values.

## 3. Results and Discussion

### 3.1. LIC_10559 Sequence Analysis Using Bioinformatics Tools and Protein Databases

The *LIC_10559* gene encodes a small protein of 181 amino acid residues with a molecular mass of 21 281.27 Da, calculated using the ProtParam tool available on the Expasy server (https://web.expasy.org/protparam, accessed on 20 February 2023). Furthermore, the search in the InterPro database indicated that LIC_10559 is an RNA polymerase σ factor that belongs to the σ^70^-family (its ECF subfamily; Sigma70_ECF—IPR039425) and contains only σ2 (aa: 33–82; similarity to InterPro: IPR007627) and σ4 (aa: 119–168; similarity to InterPro: IPR013249) domains, which are essential for interaction with the RNA polymerase core enzyme and transcription initiation (Figure 1A). In turn, Figure 1B shows the LIC_10559 structure model (3D model) obtained by using the AlphaFold Structure Database [30,31].

Further, according to the new classification of ECF σ factors, performed here for the purpose of this study at the ECF Hub web page according to Casas-Pastor et al. [32], LIC_10559 is classified into the ECF208 group and ECF208s1 subgroup. Unfortunately, the precise function of LIC_10559 as a potential ECF σ factor in leptospiral signal transduction still needs to be investigated. The amino acid sequence similarity assessed by BLAST between LIC_10559 from *L. interrogans* and *E. coli* σ^E^, which is also a member of the ECF subfamily, is 58.7%; this may suggest, but does not confirm, their similar functions. Of note, *E. coli* σ^E^ (also called σ^24^) contributes to the stress response induced by alterations in the expression and maturation of outer-membrane proteins and extreme heat shock [33]. It is important to note that LIC_10559 was found to be twofold upregulated at higher temperatures [9]; therefore, it can be speculated that this factor is implicated in the response to thermal stress occurring during infection of the host (i.e., host-induced stress).

### 3.2. Expression of LIC_10559 in E. coli pET System and Purification of Its Protein Product

Standard molecular biology methods were used for the cloning of LIC_10559 into the pET28 expression vector. This construct was generated with the use of a pair of primers (see Materials and Methods), *L. interrogans* genomic DNA as a template and PCR. As expected, the expression of the recombinant plasmid pET28*LIC_10559* in *E. coli* B21(λDE3) cells resulted in the ~23.4 kDa protein, corresponding to a 6-histidine-tagged LIC_10559 protein (Figure 2A) The molecular weight of the his_6_-tagged LIC_10559 protein (AAS69180.1; GenBank Protein Accession Number) was estimated by again using the ProtParam tool (https://web.expasy.org/protparam, accessed on 20 February 2023). The identity of overproduced LIC_10559 was confirmed by the LC-MS/MS analysis (Figure 2B). A band cut out from a Coomassie blue-stained polyacrylamide gel containing expected LIC10559 protein (Figure 2A, lane 1) was provided for this analysis. The obtained peptide map covered 89% of the amino acid sequence of LIC_10559.

Before the large-scale purification of the his_6_-tagged LIC_10559 from *E. coli* BL21 (λDE3) cells, a small-scale extraction trial was performed to determine in which subcellular fraction this protein would be found. To test the effect of temperature conditions on the solubility of this protein, bacteria overproducing LIC_10559 were grown at 28 °C or 37 °C. Cell fractionation, performed by using the CelLytic Reagent, showed that LIC_10559 was present in both—the soluble and insoluble (as inclusion bodies) forms (Figure 3). It was noticed that the temperature affected the amount of soluble protein. There was more LIC_10559 in the soluble form at 28 °C than at 37 °C (see Figure 3, lanes 6 and 2, respectively).

LIC_10559 was subsequently purified under native conditions only from the soluble fraction using immobilized metal affinity chromatography (IMAC) (Figure 4). To reduce the tendency of LIC_10559 to form inclusion bodies, the *E. coli* macro-culture for LIC_10559 separation and purification was prepared at 28 °C. A total of ~1.5 mg of the protein preparation was obtained from 2 L of the bacterial culture. The purified protein was further tested by ECL Western blotting analysis.

### 3.3. Immunogenic Potential of LIC_10559

Since it is assumed that LIC_10559 is active during infections caused by pathogenic *Leptospira* spp., we decided to examine whether sera obtained from animals infected with *Leptospira* would contain antibodies specific to LIC_10599. Activation of the host’s immune system to elicit production of specific anti-LIC_10559 antibodies could point to this protein’s involvement in the pathogenesis of leptospirosis and Leptospira pathogenicity. The immune reactivity of the recombinant LIC_10559 protein with serologically positive sera from rabbits and cattle experimentally infected with two pathogenic *Leptospira* species (*L. interrogans* and *L. borgpetersenii*) was tested by ECL Western blotting (Figure 5A,B). The bovine sera were additionally tested by an ELISA assay (Figure 6). We found that all of the tested sera prepared from *Leptospira*-infected animals, but not from the uninfected controls, strongly reacted with LIC_10559 in the Western blotting analysis (Figure 5A,B). The ELISA signals of the bovine sera from infected animals were also significantly higher than those of the uninfected controls (Figure 6; *p* < 0.0001). These results demonstrate that leptospiral infection induces the host’s humoral immune response against LIC_10559 and also indicate that LIC_10559 is an immunogenic protein. Furthermore, recognition of LIC_10559 by the humoral immune response to *Leptospira* is a strong indication that this protein is produced during infection, and hence, it may be important for leptospiral pathogenesis. Our result may support the above-mentioned speculation that LIC_10559 is active during infection.

It is worth mentioning that there are several reports that address the identification of leptospiral proteins (i.e., leptospiral antigens) recognized by the host’s humoral immune response during infection [34,35,36,37]. For this purpose, Western blotting analysis was also used in these studies. Among the proteins strongly reacting with the tested animal and human sera, a protein with a mass of 25 kDa was found in those reports. This protein has not been characterized so far, and whether it corresponds to the LIC_10599 studied here or to some other leptospiral proteins remains to be elucidated. It should be emphasized that such studies are important because the identification of leptospiral antigens appearing during infection may contribute to the development of new diagnostic and immune protective strategies.

## 4. Conclusions and Future Directions

To our knowledge, this is the first report addressing a putative ECF σ factor, LIC_10559, from the pathogenic spirochaete *L. interrogans*, and its purification and immunological properties. In this study, we used the popular *E.coli* pET expression system to overproduce LIC_10559 fused to the his_6—_tag at its N-terminus. Despite the tendency of LIC_10559 to form inclusion body aggregates during overproduction in *E. coli* cells, we were able to obtain this protein in a soluble form and test its immunoreactivity with sera from *Leptospira*-infected animals (rabbits and cattle) and healthy controls. We demonstrated that, during infection, LIC_10559 is able to activate the host’s immune system and elicit an antibody response in infected animals, suggesting its activity during leptospiral infection. Therefore, LIC_10559 could be considered in the future as a potential diagnostic antigen in the detection of leptospirosis. In addition to its use in serodiagnosis, as an active protein during infection, LIC_10559 might also have immunoprotective potential, which would be worth exploring.

Further studies and analyses are needed to better characterize LIC_10559 and determine its specific role in *Leptospira* and its effect on the host immunological responses and, most importantly, to demonstrate that this protein indeed functions as a σ factor of RNA polymerase. Our study is the first step towards determining key aspects of its function in pathogenic *Leptospira*. Certainly, the recombinant LIC_10559 produced in this study will be useful in further biochemical characterization of this protein.

## Figures and Tables

**Figure 1 pathogens-12-00512-f001:**
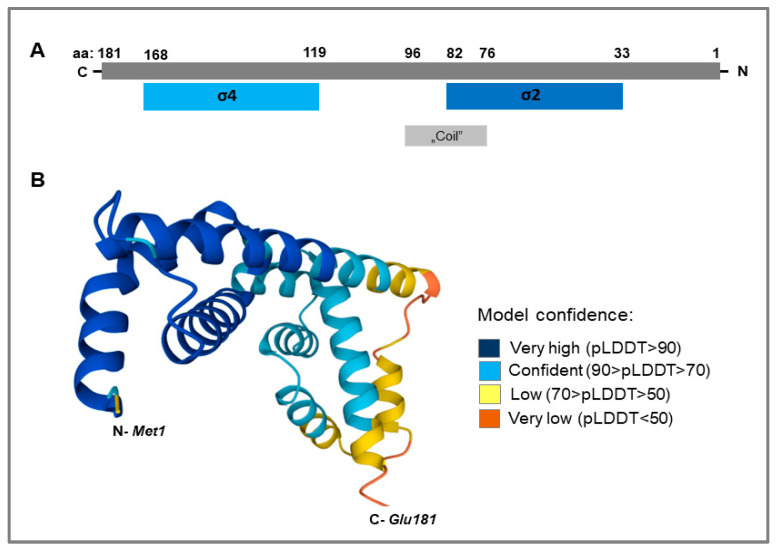
Proposed domain organization of the *L. interrogans* ECF σ factor, LIC_10559, and its 3D structure model. (**A**) The diagram shows two σ domains, i.e., σ2 and σ4, specific for the ECF σ factor subfamily indicated by the InterPro database. The “coil” motif (stabilizing and oligomerization motif in proteins) present in this σ factor, consisting of α-helices, is also indicated. (**B**) AlfaFold monomer prediction for LIC_10559 (Q72UU8- UniProt accession code). AlphaFold produces a per-residue confidence score (pLDDT) between 0 and 100. Some regions below 50 pLDDT may be unstructured in isolation. No experimental structures of LIC_10559 are available in the PDB.

**Figure 2 pathogens-12-00512-f002:**
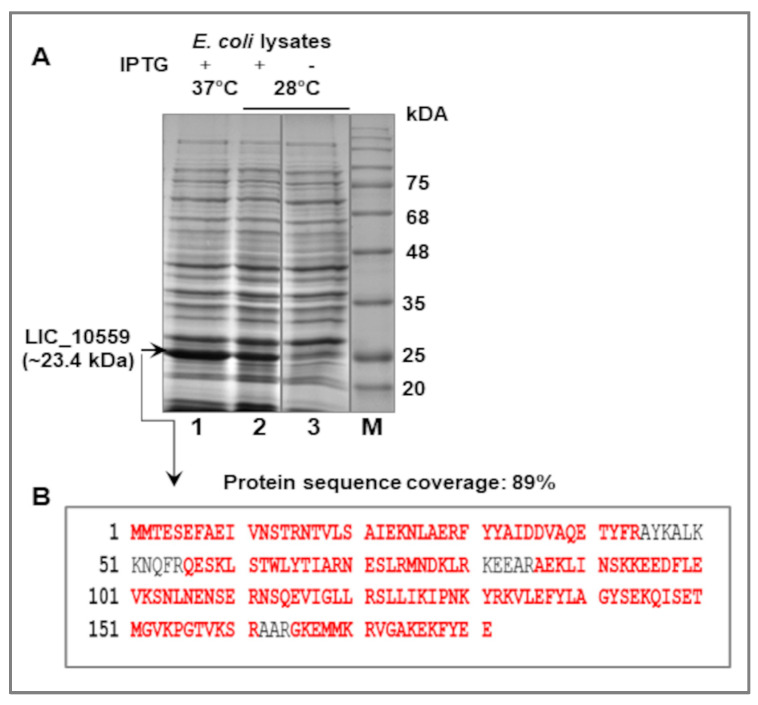
Expression of *LIC_10559* in *E. coli* BL21 (λDE3) cells and its analysis. (**A**) The Coomassie blue-stained 0.1%SDS-12.5%PAGE gel showing the lysates obtained from *E. coli* cells expressing *LIC_10559* from pET28 (at 37 and 28 °C) induced with 0.4 mM IPTG for 2 h (+) (lanes 1 and 2) and without IPTG induction (-) (lane3). Equal amounts of the lysates were loaded onto the gel. The positions of protein size markers (M; in kDa), Perfect Tricolor Protein Ladder (EURx, Gdańsk, Poland), are shown on the right. (**B**) LC-MS/MS analysis of the overproduced LIC_10559. The amino acid sequence of LIC_10559 is shown, with peptides detected by LC-MS/MS indicated in red.

**Figure 3 pathogens-12-00512-f003:**
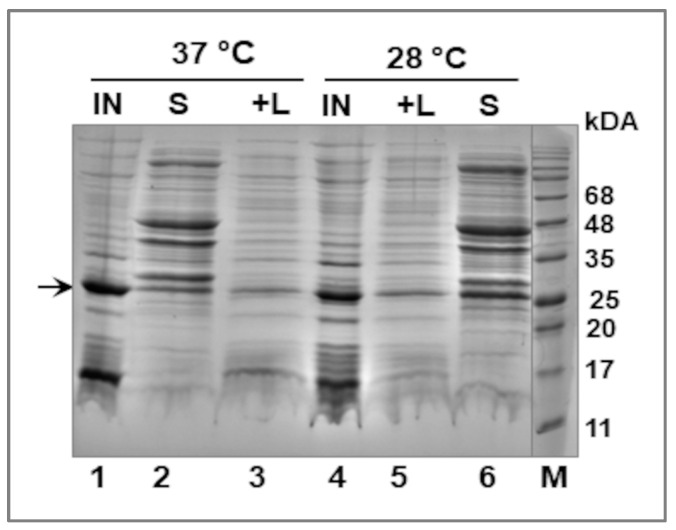
Fractionation of *E. coli* BL21 (λDE3)[pET28*LIC_10559*] cells—extraction of LIC_10559. Bacteria were grown at 28 °C or 37 °C in the presence of 0.4 mM IPTG for 2 h, and then fractionated by using the CelLytic reagent. The obtained fractions—S (soluble fraction), +L (an additional fraction obtained after lysozyme addition) and IN (insoluble fraction, containing inclusion bodies of LIC_10559)—were subsequently analyzed by 0.1%SDS-15%PAGE and Coomassie blue staining. Equal amounts of fractions were applied to the gel. The arrow indicates the position of the his_6_-tagged LIC_10559. The positions of protein size markers (M; in kDa), Perfect Tricolor Protein Ladder (EURx, Gdańsk, Poland), are shown on the right.

**Figure 4 pathogens-12-00512-f004:**
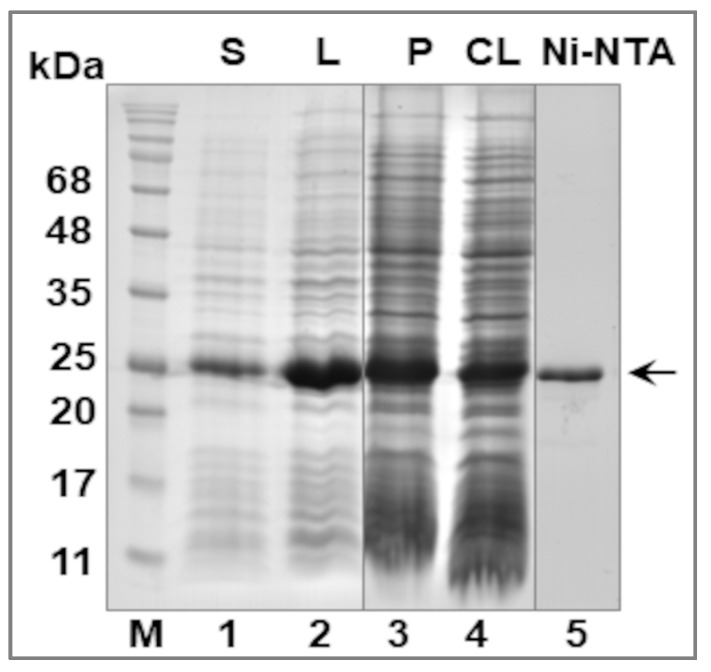
Purification of LIC_10559. Coomassie blue-stained 0.1%SDS-15%PAGE gel showing the main purification steps. Lanes 1 and 2—lysates from *E. coli* BL21 (λDE3) cells overproducing the his_6_-tagged LIC_10559 protein and grown at 28 °C, in the presence of 0.4 mM IPTG for 2 h (S, small-scale production; IPTG induction control) or 4 h (L, large-scale production; macro-culture), respectively; lane 3—pellet (P; the remaining insoluble, post-sonication and post-centrifugation material); lane 4—clarified lysate (CL) loaded onto the nickel column (Ni-NTA); lane 5—a representative fraction obtained following the Ni-NTA affinity chromatography. The arrow indicates position of LIC_10559 fused to the his_6_-tag. The positions of protein size markers (M; in kDA), Perfect Tricolor Protein Ladder (EURx, Gdańsk, Poland), are shown on the left.

**Figure 5 pathogens-12-00512-f005:**
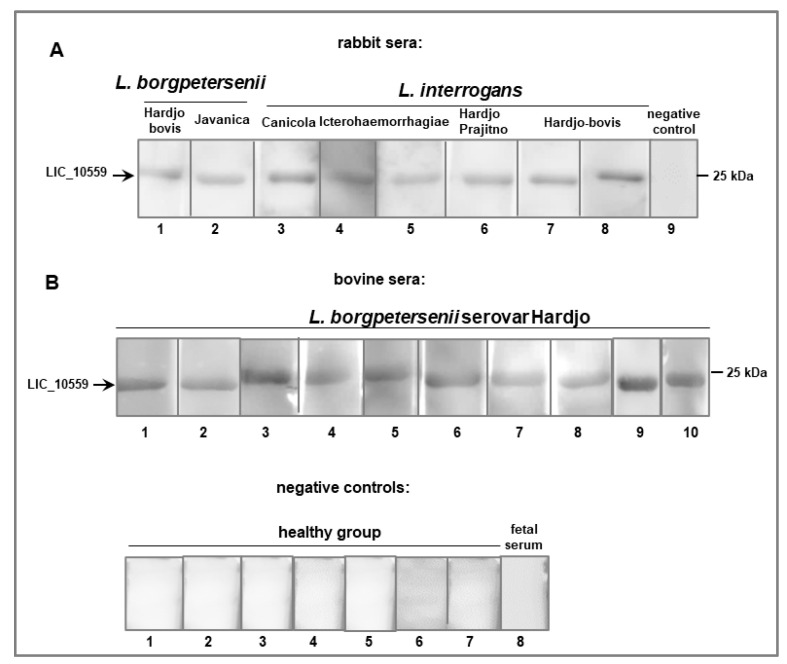
Immunoreactivity of LIC_10559 with animal sera. The purified his_6_-tagged LIC_10559 protein (100 ng) was resolved by 0.1%SDS-15%PAGE and then analyzed by ECL Western blotting using (**A**) a rabbit pre-immune control serum (a negative control), polyclonal rabbit antisera raised against *L. interrogans* and *L*. *borgpetersenii* serovars, as indicated in the figure, and (**B**) polyclonal bovine antisera raised against *L. borgpetersenii* serovar Hardjo, sera collected from uninfected cattle (uninfected healthy group; negative controls) and a fetal serum. The position of the 25 kDa protein marker (Perfect Tricolor Protein Ladder (EURx, Gdańsk, Poland) is shown on the right.

**Figure 6 pathogens-12-00512-f006:**
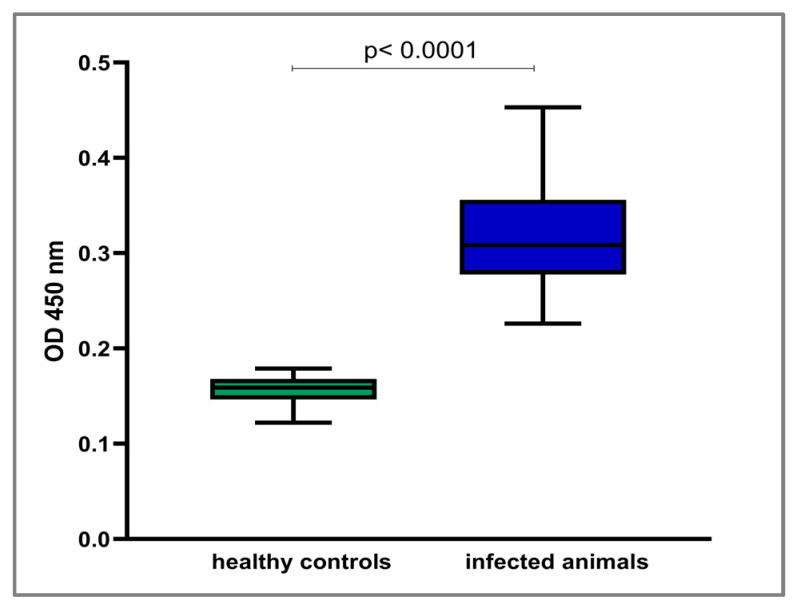
ELISA analysis of the recombinant LIC_10559 protein as a capture antigen using the abovementioned bovine sera. Fetal bovine serum was omitted in this assay. The data were analyzed using the Mann–Whitney U test. Boxes represent the 25–75% range of the data, whiskers represent min and max, and horizontal lines represent the medians.

**Table 1 pathogens-12-00512-t001:** Sigma factors predicted in *L. interrogans* (based on [17,18,25]).

Factor	Controlled Genes/Proposed Function
σ^70^ (primary σ factor)	mainly the housekeeping genes (>1000 genes)—most genes expressed during exponential growth
σ^28^	genes encoding components of the endoflagellum (*flaA1*, *flaB1*, *flaB4*) and the flagellin-specific chaperone FliS (*fliS)*; mainly cell motility
11 ECF σ^E^-type factors	extracytoplasmic function (469 putative binding sites in the *L. interrogans* genome for all ECF σs); heat shock response, stress survival and virulence (*clpB*)
σ^54^	genes encoding putative lipoprotein and the ammonium transporter AmtB; mainly control of internal concentration of ammonia and environmental adaptation

## Data Availability

The DNA sequence of the *L. interrogans LIC_10559* gene was retrieved from GenBank at the NCBI website (accession number NC_005823). The protein sequences of the *L. interrogans* LIC_10559 and *E. coli* σ^E^ proteins were retrieved from UniProtKB (accession number Q72UU8; Q72UU8_LEPIC/https://www.uniprot.org/uniprotkb/Q72UU8/entry and P0AGB6 (RPOE_ECOLI)/https://www.uniprot.org/uniprotkb/P0AGB6/entry, accessed on 20 February 2023.

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
