# Peer review of "Immunoreactivity of a Putative ECF σ Factor, LIC_10559, from Leptospira interrogans with Sera from Leptospira-Infected Animals"

_pathogens, 2023, doi:10.3390/pathogens12040512_

Round 1

Reviewer 1 Report

The authors studied an exciting and vital subject in molecular biology, parasitology, and medicine. Undoubtedly, the findings on the putative ECFs factor, LIC_10559, from Leptospira interrogans are of scientific interest to the readers of the pathogens journal. My only comment (but highly important for the interpretation) is on the quality of the figures showing the results of SDS-PAGE and Western blot. The lack of quality and cleanliness of the figures detracts from their interpretation. Please take as an example the following manuscripts to improve yours.

Protein J. 2013 Oct;32(7):585-92. doi: 10.1007/s10930-013-9518-x. See fig. 3

Pathogens. 2022 Sep 17;11(9):1059. doi: 10.3390/pathogens11091059. See fig. 2

Besides, the archive is supposed to find the original non-edited blots and gels containing the edited ones.

Please include both to get the quality of your findings better.

Author Response

Reviewer 1:

Undoubtedly, the findings on the putative ECFs factor, LIC_10559, from Leptospira interrogans are of scientific interest to the readers of the pathogens journal. My only comment (but highly important for the interpretation) is on the quality of the figures showing the results of SDS-PAGE and Western blot. The lack of quality and cleanliness of the figures detracts from their interpretation. Please take as an example the following manuscripts to improve yours.

Protein J. 2013 Oct;32(7):585-92. doi: 10.1007/s10930-013-9518-x. See fig. 3

Pathogens. 2022 Sep 17;11(9):1059. doi: 10.3390/pathogens11091059. See fig. 2

Besides, the archive is supposed to find the original non-edited blots and gels containing the edited ones.

Please include both to get the quality of your findings better.

Response:

We are grateful to the Reviewer for their insightful reviews and taking the time to write a review.

We would like to emphasize that the original photos/figures obtained using the Azure imaging system are included in our manuscript. Figures/photos were described in the Power Point program, and then adapted to the journal's requirement, i.e. minimum 1000 pixels width/height, or a resolution of 300 dpi or higher, using GIMP. LIC_10559 is clearly visible on all gels and blots. Therefore, we are surprised by the Reviewer's comment.

It's worth mentioning that the blotting membranes were trimmed between the 35- and 17- kDa protein markers to limit the use of valuable animal sera. Then, the membranes were cut into small strips which were then separately incubated with various animal sera and developed using BioVision ECL substrates, and imaged using the Azure imaging system (Azure Biosystems). Certainly, the quality of blots incubated with animal sera will not be comparable to the quality of blots incubated with a specific antibody (e.g. mAbs). In addition, small strips of membranes are more difficult to handle than whole membranes, especially during washing and incubation with secondary antibody solution. Small blot strips can stick together, which can result in a larger blot background. The blots were repeated so that the best strips could be selected for our manuscript. Despite these difficulties, the LIC_10559 is detectable on all membrane strips (after incubation with sera of Leptospira- infected animals).

As suggested by the Reviewer, we attach original photos/figures obtained directly from the Azure imaging system, without a description and arranging the blotting strips in a specific order. 

Reviewer 2 Report

This can best be described as a preliminary study to understand leptospirosis immunobiology. They have been able to successfully clone and express leptospiral protein and subsequently showed that the protein triggers humoral immunity in rabbits and cattle. LIC_10559 is well known immunogenic protein which is present in pathogenic isolates. Thus, it is expected to initiate immune responses. I am not sure about the novelty part!

The authors need to carry forward this study by doing simple immunological experiments: cytokine profiling, cell proliferation assay, or immunization-challenge assays with the protein to get the paper accepted.

Author Response

Reviewer 2:

This can best be described as a preliminary study to understand leptospirosis immunobiology. They have been able to successfully clone and express leptospiral protein and subsequently showed that the protein triggers humoral immunity in rabbits and cattle. LIC_10559 is well known immunogenic protein which is present in pathogenic isolates. Thus, it is expected to initiate immune responses. I am not sure about the novelty part!

The authors need to carry forward this study by doing simple immunological experiments: cytokine profiling, cell proliferation assay, or immunization-challenge assays with the protein to get the paper accepted.

Response:

We are grateful to the Reviewer for their insightful reviews and taking the time to write a review.

During the preparation of our manuscript, we did not find any report that would address an immunogenic potential of LIC_10559 - a putative ECF σ factor from L. interrogans. We would be grateful to the Reviewer for pointing out such a report. So that it could be discussed in our manuscript. Until now, we have been convinced that our research is the first approach that demonstrates the immunoreactivity of LIC10559 with sera of Leptospira-infected animals and its immunogenic potential.

In our opinion, the Reviewer's suggestion to perform long-term immunological experiments, including the cytokine profile, cell proliferation assay, and immunization-challenge assays go beyond the scope of this study. Certainly, this is a very interesting experimental proposal that can be carried out in the future.

Round 2

Reviewer 1 Report

Suggestions should have been considered more.

The manuscript and figures are the same as the first version. The current version remained the same. The authors did not include the non-edited figures. Nonetheless, SDS-PAGEs from figures 2 and 4 are edited from the original gels. The authors had to show the non-edited gels. The author's justification for blots is invalid for the western blot using anti-HisTag (fig. 2B). They still need to include the complete membrane of this assay.

Author Response

Reviewer 1:

The manuscript and figures are the same as the first version. The current version remained the same. The authors did not include the non-edited figures. Nonetheless, SDS-PAGEs from figures 2 and 4 are edited from the original gels. The authors had to show the non-edited gels. The author's justification for blots is invalid for the western blot using anti-HisTag (fig. 2B). They still need to include the complete membrane of this assay.

Response:

Once again, thank you very much to the Reviewer for taking the time to write our review.

We tried to replace the photos/figures with the originals. Figures showing SDS-PAGE gels have been replaced with new onces, i.e. non-edited figures. Since I am not able to repeat the Western blotting with anti-His-tag antibody (I am on sick leave from August 1st 2022 to July 31st , 2023) to show the complete membrane of this assay, I have decided to remove panel B (Figure 2) showing the anti-His-tag blot in the previous version of our manuscript. Removal of this panel has no significant impact on the further results of our studies. In addition, we showed that LIC_10559 can be purified by affinity chromatography (IMAC), which proves the presence of the his6-tag at its N-terminus. Thus, in the revised version of the manuscript, Figure 2 consists of two panels A and B. Panel A shows the Coomassie blue-stained 0.1%SDS-12.5%PAGE gel, and panel B demonstrates LC-MS/MS analysis of the overproduced LIC_10559. We hope this is a good solution for this problem. Therefore, minor changes were also made to the text of the manuscript (lanes: 187-190, 248, 252 and 261-264).

Reviewer 2 Report

The functional characterization is needed for the acceptance of this paper. 

Author Response

Reviewer 2:

The functional characterization is needed for the acceptance of this paper.

Response:

Once again, thank you very much to the Reviewer for taking the time to write our review.

We agree with the Reviewer that the functional characterization is an important step in research focusing on LIC_10559. However, it goes beyond the scope of this study. We have been working on the functional characterization for over 2 years. However discovering a function of LIC_10559  is a huge challenge that will take at least another 2 years. Determining the function of LIC_10559 is associated, among others, with the construction of the appropriate Leptospira mutant, which is not a simple research task due to the limitations of modern genetic tools available for pathogenic Leptospira spp. We have planned the construction of such a mutant in our research project that is currently under evaluation. We are going to generate L. interrogans LIC_10559 mutant by random (using Himar1 transposon) or targeted (allelic exchange) mutagenesis. We are also considering the use of CRISPRi for gene silencing. Construction of this mutant and its complemented strains is going to be performed in collaboration with Prof. M. Picardeau (Institut Pasteur) who has extensive experience with L. interrogans mutagenesis. We have also planned in our project other experiments which will introduce us to functions of LIC_10559. Once again, We would like to emphasize that the functional characterization of LIC_10559 proposed by the Reviewer is a huge and long-term challenge. We would like to emphasize that our present study is the first step towards determining key aspects of its function in pathogenic Leptospira. This manuscript will be the first report addressing a putative ECF σ factor, LIC_10559 from the pathogenic spirochaete L. interrogans. We demonstrated that during infection, LIC_10559 is able to activate the host’s immune system and elicit antibody response in infected animals, suggesting its activity during leptospiral infection. This is an important finding.  

Round 3

Reviewer 1 Report

You have to be more careful in the future concerning quality of your images an results.

Reviewer 2 Report

I can understand the problem authors are facing. In the present situation nothing more can be done hence  we can accept the ms for publication.